# A Taxonomy of Failing Bug Reproduction Tests on SWT-Bench

## Abstract

Automatic Program Repair (APR) tools require high-quality test generation models to validate candidate patches. Despite the advancements of contemporary automated bug reproduction testing tools in resolving a greater quantity of issues, they continue to struggle with issues of higher complexity. These struggles may, in part, be caused by a framework's general lack of problem-specific context. Furthermore, the reasons why these frameworks fail to resolve issues have never been categorized. This paper fundamentally contributes to resolving this issue by providing a taxonomy of bug reproduction test failures. This study makes use of the AssertFlip and OpenHands frameworks, the two highest-scoring open-source tools on the SWT-Bench Verified leaderboard. The significance of model selection on the causes of failure was also explored by utilizing GPT-5-mini and GPT-4o-mini on both tools. The error traces of these frameworks were then categorized to determine granular modes of failure. Through this process, the most prominent causes of failure included mechanical and implementation failures.

Overall, the tool's generation approach has a significant impact on the precision and coverage, while the selection of the model dramatically alters the quality and quantity of generated tests. Specifically, more powerful models with higher reasoning capability tended to present fewer failures due to a lack of technical or problem-specific context. Our experiment also shows evidence that our taxonomy can be used as concrete guidance to augment the issue description to improve the overall generation quality.

**ACM Reference Format:**
Anonymous Author(s). 2018. A Taxonomy of Failing Bug Reproduction Tests on SWT-Bench . In *Proceedings of Make sure to enter the correct conference title from your rights confirmation email (Conference acronym 'XX)*. ACM, New York, NY, USA, 10 pages. https://doi.org/XXXXXXX.XXXXXXX

## 1 Introduction

The costly nature of program repair spurred the development of the Automated Program Repair (APR) subfield. APR tools automatically generate patches for software bugs when provided with an issue description, ground-truth tests, and a buggy codebase [16, 20, 26, 39, 40, 43]. The effectiveness of these tools depends on the availability and quality of comprehensive contextual information, particularly high-quality bug-reproduction tests that validate candidate patches. To address this dependency, automated bug reproduction test generation has emerged as a complementary research area, providing the necessary test suites that APR

systems require. Recent advances in Large Language Models for Software Engineering (LLM4SE) have catalyzed progress in this domain, leveraging the capacity of LLMs to both generate and evaluate test cases. Open-source frameworks such as AssertFlip [18] and OpenHands[36] exemplify this trend, enabling researchers to generate bug-reproduction tests at scale through locally executable, LLM-driven approaches.

Despite these advancements, current state-of-the-art LLM-driven bug reproduction frameworks exhibit substantial limitations when faced with complex software issues, particularly those requiring deep comprehension of external library semantics and intricate API interactions. Empirical evidence from the SWT-Bench Verified benchmark reveals that even the highest-performing open-source frameworks, AssertFlip (45.5% resolution rate) and OpenHands with GPT-5-mini (62.4% resolution rate), ranking third and fifth respectively on the leaderboard [21], fail to reproduce more than one-third of the test cases. More importantly, the underlying causes of these failures remain uncharacterized in existing literature. Without a systematic understanding of failure patterns, practitioners cannot effectively diagnose weaknesses in their approaches, nor can researchers identify targeted interventions to advance the state of the art. A comprehensive taxonomy of bug reproduction test failures would address this gap by providing: (1) a structured framework for analyzing tool limitations, (2) insights into how tool choices and model selection influence failure patterns, and (3) an evidence base for developing targeted mitigation strategies. Such a taxonomy is essential for transitioning bug reproduction research from incremental performance improvements to principled, failure-informed design.

This study addresses these gaps through a systematic empirical analysis of bug reproduction test failures across multiple dimensions. We examine two leading open-source frameworks, AssertFlip and OpenHands. To isolate the effects of model selection on failure patterns, we evaluate the frameworks using GPT-4o-mini and GPT-5-mini. This controlled experimental design enables us to disentangle tool-specific limitations from model-dependent weaknesses. We analyze 254 failing test instances across all tool-model configurations and manually categorize each failure. Our granular analysis reveals not only which failure patterns dominate, but also how the distribution of these patterns shifts as a function of tool architecture and model reasoning capacity. In addition, we investigate whether simple template-based augmentation guided by our taxonomy can remediate the identified failure patterns. Specifically, we apply a targeted method to a representative sample of failing instances, augmenting issue descriptions with failure-class-specific guidance derived from our taxonomy.

This paper fundamentally seeks to provide insights on the common causes of failure within open source bug reproduction frameworks. We particularly seek to analyze the effect of tool selection and model selection in determining the prevalence of specific bug classes. Given these two objectives, the following research questions can be formulated:

- **RQ1**: What are the primary failure modes of LLM-based bug reproduction tools?
- **RQ2**: How does the tool generation approach affect failure class distributions?
- **RQ3**: How model's knowledge and reasoning ability alter failure class distributions?

**Contributions.**This work makes the following contributions to automated bug reproduction research:

- **A comprehensive taxonomy of bug reproduction test failures:** We develop a new failure taxonomy to capture the specific failure mode of LLM-based bug reproduction, introducing fine-grained subclasses to enable deeper analysis. Our taxonomy provides a standardized vocabulary for characterizing 254 real-world failures across state-of-the-art frameworks.
- **Empirical analysis of failure pattern distributions:** We quantify how tool architecture and model selection influence failure distributions, revealing that tool choice can affect the precision and coverage of test generation and model reasoning capability significantly impacts the quantity and quality of generated tests.
- **Evidence for taxonomy-driven remediation:** We demonstrate that targeted remediation strategies are plausible with the help of our taxonomy. Applying to previously failing instances, 7 out of 11 augmentation attempts successfully improved the generated test. This indicates that with our taxonomy guidance, practitioners would have a concrete remediation method without much effort.

## 2 Background

This section reviews essential background on bug reproduction testing and benchmarking practices.

### 2.1 LLM-Driven Bug Reproduction Testing

Bug reproduction testing aims to automatically generate test cases that expose reported software defects, and it is a key prerequisite for automated program repair and regression testing [45]. A bug-reproducing test must satisfy two essential properties: it fails on the buggy version and passes after the fix, thereby serving as an executable oracle for both defect presence and patch correctness [45]. In practice, the main source of problem-specific context is the bug report, yet producing reliable reproductions manually is labor-intensive and often deferred or ignored by developers [8, 30]. This motivates automated reproduction pipelines that translate issue context into runnable tests. Earlier work explored automated crash reproduction strategies that leverage partial signals such as crash traces to recover triggering executions [30]. More recently, LLM-driven frameworks generalize beyond crash-only settings by synthesizing reproduction tests directly from natural language issue descriptions and iterative execution feedback, enabling reproduction at scale [17, 18]. A central challenge in this setting is not only generating test code, but also constructing *correct* assertions (oracles) that align with the issue's expected behavior. CodeT addresses this by validating candidates through cross-sample agreement using a RANSAC-inspired consensus mechanism to filter inconsistent tests [9]. Complementarily, semantic enrichment methods such as TELPA [41] leverage counterexamples to steer the model toward deeper system-under-test understanding. AssertFlip [18] operationalizes oracle construction in the reproduction setting by generating a passing test from an LLM-produced plan.

### 2.2 Benchmarking

Benchmarking provides standardized evaluation protocols for systematic comparison of competing systems under controlled conditions [5, 6, 11, 19, 24]. Historical benchmarks have enabled quantitative assessment across diverse domains, including LAN networks, database management systems, and information extraction [29]. The advances in machine learning has motivated domain-specific benchmarks tailored to particular evaluation objectives. In the security domain, TDDBench [44] assesses methods for detecting training data exposure. For software development, BaxBench [33] evaluates automated backend generation systems on functionality and security criteria. In the LLM4SE domain, SWE-Bench [16] establishes evaluation baselines using real-world software engineering issues.

*2.2.1 Reproduction Test Benchmarks.* Automated bug reproduction test generation requires standardized benchmarks for evaluation. SWT-Bench [21] addresses this need by adapting problems from SWE-Bench to create a bug reproduction benchmark. Git-BugJava [28] provides benchmarking functionality for Java-based bug reproduction, comprising 199 bugs with patches from 55 open-source repositories. To broaden coverage beyond Python/Java issue-to-test settings, CrashJS [25] contributes a benchmark of 453 Node.js crashes for automated crash reproduction, while BugsInDLLs [23] curates 112 reproducible environments across JAX, TensorFlow, and PyTorch to enable systematic evaluation of testing techniques against deep-learning-library bugs. For lower-level crash scenarios, Live-kBench [15] further provides an evolving Linux kernel crash-resolution benchmark.

## 3 Related Work

This section reviews existing taxonomies for test failure classification and state-of-the-art frameworks that underpin our study design and taxonomy construction.

### 3.1 Taxonomy

*3.1.1 Taxonomy in Software Testing.* Taxonomies establish standardized classification schemes that enable systematic analysis and comparison across studies in software testing [12, 32, 34, 37]. Beizer's seminal defect taxonomy introduced a foundational classification framework comprising nine distinct fault categories [7]. Subsequent taxonomies have adapted and extended this foundation to address evolving testing paradigms. Felderer and Beer refined Beizer's categories to better capture failures in requirements-based testing contexts [13]. Beyond defect classification, taxonomies have been developed to categorize testing methodologies themselves, as exemplified by Shaukat et al.'s systematic classification of testing techniques [27].

Taherkhani et al. [31] presents a taxonomy of LLM-generated test failures comprising seven invalidity categories. We take inspiration, although with a different target subject and definition set, from this taxonomy.

## 3.2 Automated Reproduction Test Generation

Automated bug reproduction test generation approaches fall into two categories. Agentic approaches prioritize planning capabilities and task autonomy at higher computational cost. Agentless approaches offer lower-cost alternatives that, as demonstrated by Xia et al. [38], achieve competitive effectiveness.

Several agentic approaches have been developed for bug reproduction test generation. SWE-Agent [42] introduced a software-engineering-specific agentic system with an enriched user interface. OpenHands [36] ranks among the most effective agentic systems on SWT-Bench, extending SWE-Agent through multi-capability agent deployment to provide a more generalized framework. AEGIS [35] represents the first domain-specific agent for bug reproduction, incorporating a text generation module followed by an FSM-based feedback module. BRT Agent [10] augments the established, LI-BRO [17] framework by integrating a reasoning module into its reproduction pipeline. LIBRO [17] represents an early agentless framework that combines few-shot prompting with post-generation ranking. Otter [3] employs LLM-driven planning instead of ranking, sequentially localizing files, generating a resolution plan, revising it, and executing. e-Otter++ [4] extends this approach by generating multiple candidate tests using Otter and critiquing and repairing them via LLM feedback. Issue2Test [22] advances SWT-Bench performance over SWE-Agent, and LIBRO by incorporating an error categorization step to refine failing tests. AssertFlip [18] adopts a similar refinement strategy through assertion inversion by generating a passing test from an LLM-produced test plan, flipping the assertion, and validating through LLM-based criticism.

## 4 Study Design

This study focuses on investigating the common pitfalls of automatic reproduction test generation and how our findings can help practitioners utilize state-of-the-art LLM-based generation tools. This study was conducted with a selection of two top open-source generation frameworks on the SWT-Bench Verified leaderboard [21]. We also perform our study on two general-purpose and lightweight LLMs, GPT-4o-mini and GPT-5-mini, which represent typical usage in the real world. Finally, based on the existing taxonomy [14], we expand and label the invalidity of generated reproduction tests. The SWT Bench evaluation harness is used as a uniform evaluation method for each tool-model pair. There may be issues in each pair which cannot be assessed as they do not generate a test for the harness to evaluate. For instance, AssertFlip with GPT-5-mini may have more labelled failures than AssertFlip on GPT-4o-mini as the latter was not able to generate the same quantity of inferences for the SWT Bench harness to evaluate. Finally, to test the practicality of our findings, we apply some concrete insights from our taxonomy to improve the test generation from the practitioner's point of view and observe significant improvement with simple issue description augmentation.

### 4.1 Study Subjects

Several reproduction test generation frameworks are reported on the SWT-Bench leaderboard; however, we focus on open-source frameworks to aid with taxonomy construction and reasoning about failure modes as closed-source frameworks would hinder our ability to closely investigate the root causes of generation failures. Assert-Flip and OpenHands represent the top two open-source frameworks on the SWT-Bench Verified leaderboard. Even if AssertFlip and OpenHands represent weaker frameworks overall, it does not hinder our study, as we focus on investigating modes of failure and how one could improve the performance from the practitioner's point of view.

In this study, GPT-4o-mini and GPT-5-mini are chosen for their lightweight reasoning capability. While both models are marketed for their lightweight nature and reasoning capabilities, it should be noted that GPT-5-mini outperforms GPT-4o-mini [1]. The dichotomy between these two models enables the taxonomy results to be analyzed on the basis of model as well as the generation frameworks. This further analysis will provide insights on the model's influence in terms of generated reproduction test failure.

### 4.2 Taxonomy Construction

In general, taxonomization aids in categorizing failing tests by using a standard language. Individual test failure taxonomies were decided upon after reviewing the available existing taxonomies in the background and related work.

As existing taxonomy categories were broad for our subject matter, we decided to develop our own. This synthesis enables for a more detailed analysis of test generation failures and provides more concrete and detailed insights so that practitioners can improve the generated test with targeted issue description augmentation.

The construction of this taxonomy required three of the authors to participate in the labelling process. Two authors began the process by agreeing on an initial set of categories and independently labelling all failing instances from our selected frameworks and LLMs. For this stage, our inter-annotator agreement, computed with Cohen's Kappa coefficient, is **0.953**, indicating a good agreement between annotators. In the event of a labelling disagreement, a third author acted as a tie-breaker to ensure every label receives a majority consensus.

Table 1 shows our full taxonomy with subcategories along with their definitions and examples. We present the *Misimplementations*, *Mechanical Failure* and *Requirement Misunderstanding* classes to respectively represent instances of improper test state, external failure, and an incapability to understand the requirements of an issue. *Misimplementations* contains the *Logical Failure*, *Incorrect Assertion*, and *Incorrect Input / Mock* subclasses to distinguish the specific manner by which a test fails to generate a reflective bug reproduction test. Additionally, *Mechanical Failure* is expanded upon by including the *Wrong API Call*, *Output Format Inconsistency*, *Environment Error*, *Not Implemented*, and *Incorrect File Reference* subclasses to represent instances of incorrect library interaction, failed environment setup, and invalid test generation. Lastly, Requirement Misunderstanding includes the *Misunderstanding Edge Case Logic* and *Misunderstanding Function Logic From Natural Language* subclasses to articulate the source of a tool's inability to internalize a problem's requirements.

**Table 1: Definitions and Examples of Every Taxonomy Class and Subclass**

| Class | Subclass | Definition | Example |
|---|---|---|---|
| Mechanical Failure | Not Implemented | A test is not provided by the tool | *#TODO 1. mock service:.* |
| | Output Format Inconsistency | A test, despite being logically correct, expects an output to be formatted differently from its actual format | `assert ('axis', 'azimuth_time') == ('azimuth_time', 'axis')` |
| | Environment Error | An error within the test environment occurs prior to execution | pytest.py: error: unrecognized arguments: –no-header |
| | Incorrect File Reference | The generated test references an incorrect file path | ~/my_file.py does not exist |
| | Wrong API Call | Incorrect interaction with an external library | Treating an SQL table as a simple string |
| Misimplementations | Incorrect Input/Mock | The test fails to provide correct input or mock input values to reproduce the buggy state | Incorrectly mocking a key leads to a hashing discrepancy |
| | Incorrect Assertion | The oracle checks the wrong variable or state | Asserting the opposite state to a given issue |
| | Logical Failure | The test case executes a sequence of operations that is syntactically correct but semantically disconnected from the bug's trigger conditions | Monkeypatching Axes3D.draw to detect invisibility logic rather than verifying the rendered output layers. |
| Requirement Misunderstanding | Misunderstanding Edge Case Logic | A test does not sufficiently acknowledge or address an edge case within its logic | Does not account for a shared reference being a shallow copy |
| | Misunderstanding Function Logic From Natural Language | A test fails to sufficiently interpret natural language logic | Incorrect translation of non Latin characters results in a downstream DNS Translation error |

## 5 Experiment Setup

The test generation frameworks are executed on a 2020 MacBookPro with 32 GB of memory running macOS 15.6.1. Following the replication instructions on the respective GitHub repositories for AssertFlip [18] and OpenHands [2], Docker with Anaconda is configured. To minimize resource wastage, the experiment on OpenHands with GPT-5-mini was reused from prior work [21]. The hyperparameters, such as the number of generation iterations, for both AssertFlip and OpenHands are kept at the default values to ensure consistent results.

The SWT-Bench evaluation harness [21] was used to assess the generated tests from each tool-model pair. The failing tests, along with their associated error traces (the trace of a test's execution) from the evaluation harness, were then extracted to label and construct the taxonomy.

## 6 Results

In this section, we provide the investigation and answer for each research question. Specifically, we present the overall distribution of failure classes and discuss the common failure patterns to answer RQ1. Next, we analyze the differences between frameworks and LLMs and discuss the specific failure modes when different frameworks and LLMs are used. And finally, we present some concrete experiments showing how the finding from our taxonomy can help practitioners make concrete arguments to test generation frameworks and improve their performance.

### 6.1 RQ1: What Are the Primary Failure Modes of LLM-Based Bug Reproduction Tools?

To answer this research question, three authors manually reviewed the generated reproduction test failures and labeled each instance with one of the categories from our taxonomy presented in table 1. This taxonomy was constructed concurrently with labeling to ensure consistency and completeness.

Figure 1.a shows the distributions of the reproduction test failures across all three framework-LLM combinations. Each bar represents a failure class with texturing to represent failure subclasses. The distribution is presented as the percentage of the particular class with respect to the total number of generated test cases. This will show the relative distribution between different tool-model combinations. However, since each configuration would have a different number of generated test cases, table 2 also shows the number of failed tests in each failure class, as well as the number of valid tests and the total number of generated tests.

Overall, the most prevalent failure classes across the 254 test failures are *Mechanical Failure* and *Misimplementations*. These two classes were observed in 30.0% of all generated tests. The three most prevalent subclasses of *Mechanical Failure* include *Wrong API Call* (8.1%), *Output Format Inconsistency* (2.3%), and *Environment Error* (4.9%). The latter subclass presents an interesting mode of failure by which a test cannot be executed due to an error in establishing its runtime environment. Furthermore, the dominance of *Wrong API Call* highlights the detriment of a lack of library-specific context. For instance, *sympy-15875* failed as the model did not assume that the method under test could return *None*. Therefore, a relatively

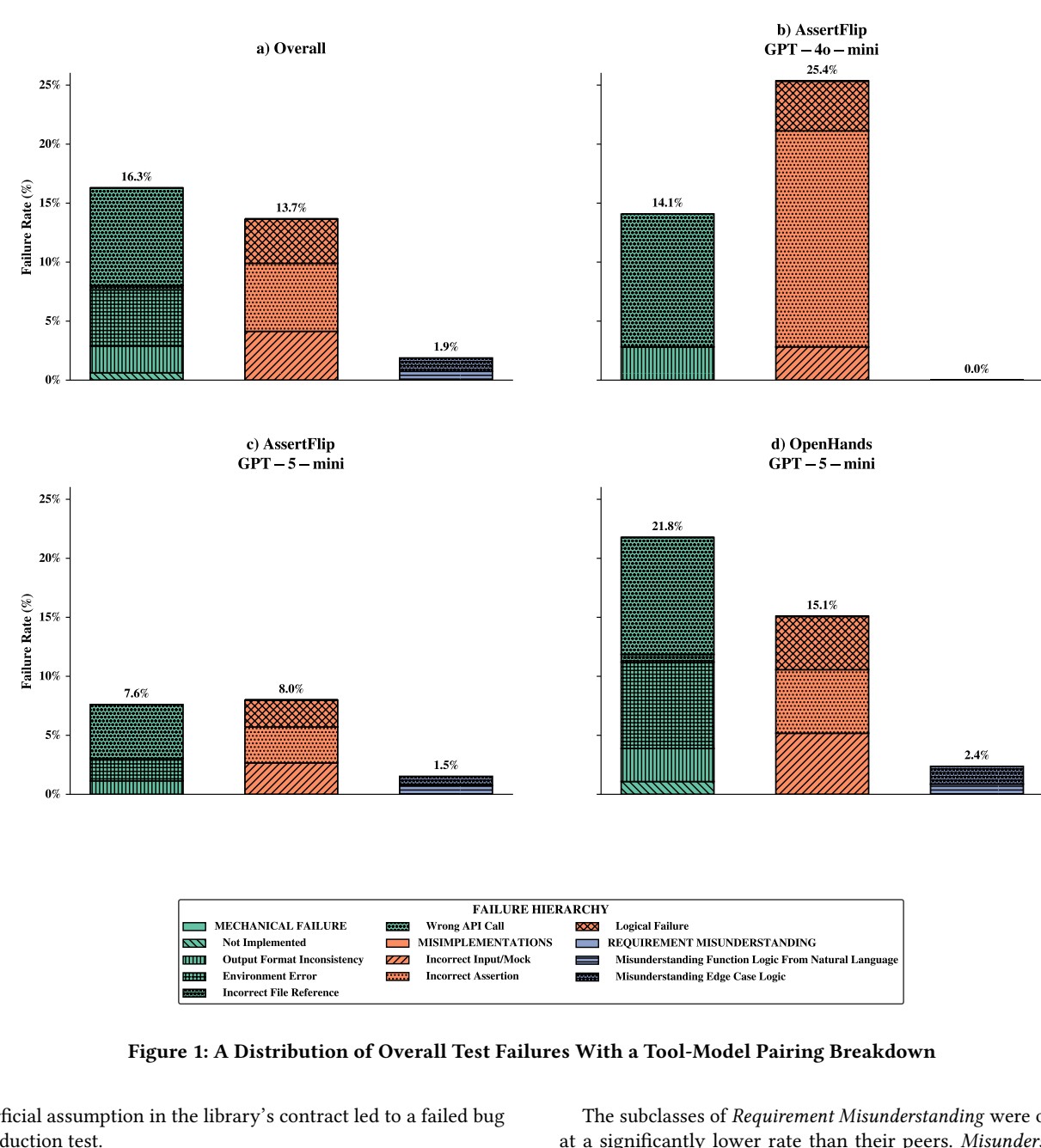

**Figure 1: A Distribution of Overall Test Failures With a Tool-Model Pairing Breakdown**

superficial assumption in the library's contract led to a failed bug reproduction test.

The three subclasses in *Misimplementation* include *Incorrect Assertion* (5.8%), *Incorrect Input/Mock* (4.2%) and *Logical Failure* (3.7%). These subcategories represent the main challenges of test generation: how to generate (1) a valid test oracle with (2) appropriate input, with (3) the correct testing logic. For instance, in the issue *astropy-14309*, the model assumed that resolving the given Index-Error implied asserting an empty file registry despite there being other file types within the issue, thus resulting in an *Incorrect Assertion*. This significant frequency demonstrates that when the tool fails to generate an effective test, it typically does so for superficial contextual or logical reasons.

The subclasses of *Requirement Misunderstanding* were observed at a significantly lower rate than their peers. *Misunderstanding Edge Case Logic* and *Misunderstanding Function Logic From Natural Language* were respectively observed at a rate of 1.1% and 0.8%. These failures, unlike the most common subclasses, typically include added logical complexity within their class-specific domain. For instance, a non-Latin character translation failure demonstrates the added complexity of *Misunderstanding Function Logic From Natural Language* as it requires domain knowledge on non-Latin alphabets and Domain Name Service (DNS) translation.

Overall, the majority of overall failing issues tend to fail as a result of poor test implementation or a lack of library context. Conversely, the least observed failure classes tend to suffer from a

**Table 2: Raw Failure Distributions in Each Tool-Model Pairing and Overall**

| Class | Subclass | Overall | AssertFlip GPT-4o-mini | AssertFlip GPT-5-Mini | OpenHands GPT-5-mini |
|---|---|---|---|---|---|
| Mechanical Failure | Not Implemented | 5 | 0 | 0 | 5 |
| | Output Format Inconsistency | 19 | 2 | 4 | 13 |
| | Environment Error | 39 | 0 | 5 | 34 |
| | Incorrect File Reference | 3 | 0 | 0 | 3 |
| | Wrong API Call | 65 | 8 | 11 | 46 |
| | *Class Total* | 131 | 10 | 20 | 101 |
| Misimplementation | Incorrect Input/Mock | 33 | 2 | 7 | 24 |
| | Incorrect Assertion | 46 | 13 | 8 | 25 |
| | Logical Failure | 29 | 3 | 6 | 21 |
| | *Class Total* | 109 | 18 | 21 | 70 |
| Requirement Misunderstanding | Misunderstanding Function Logic | 6 | 0 | 2 | 4 |
| | Misunderstanding Edge Case Logic | 9 | 0 | 2 | 7 |
| | *Class Total* | 15 | 0 | 4 | 11 |
| **Failing Tests** | | 254 | 28 | 45 | 182 |
| **Generated Tests** | | 798 | 71 | 263 | 464 |

higher logical complexity. This finding suggests that most issues fail as a result of a lack of logical or contextual clarity rather than a high logical complexity, and further clarification in the issue text would help the model generate more correct test cases. We will explore hypotheses further in Section 7.

> **Finding 1:** *Mechanical Failure* and *Misimplementations* are the most prevalent failures. These classes typically fail due to superficial logical or contextual assumptions, whereas the least common failure classes tend to fail on account of their more complex logic.

## 6.2 RQ2: How Does the Tool Generation Approach Affect Failure Class Distributions?

In this RQ, we seek to determine the influence of a tool's test generation strategy on the distribution of observed failure classes. To do so, we investigate the differences in frequency and proportion of observed failures between AssertFlip's 263 generated tests (figure 1.c) and OpenHands' 464 generated tests (figure 1.d) on the same LLM (GPT-5-mini). This analysis would provide more insight into the differences between these two tools.

The overall per-tool distributions differ significantly in their top two observed classes. For instance, *Misimplementations* (8.0% vs. 15.1%) and *Mechanical Failure* (7.6% vs. 21.8%) go from being relatively equal on AssertFlip to significantly imbalanced on OpenHands, with *Requirement Misunderstanding* remaining relatively marginal. This trend is primarily driven by the high quantity of *Environment Error* on Openhands-GPT-5-mini. Due to its agentic approach, OpenHands might misconfigure the test, leading to missing files and incorrect environmental setup errors.

Intriguingly, the relative proportion in *Misimplementations* remains stable across both tools. For instance, *Incorrect Assertion* (3.0%) on AssertFlip and OpenHands (5.4%) comprise approximately

a third of their *Misimplementations*. This proportion suggests a failure parity; despite different operational paradigms, AssertFlip's specialized inversion versus OpenHands' agentic interaction, both tools reach a similar bottleneck where implementation failure presents itself at nearly identical rates.

While both tools reach similar bottlenecks, OpenHands produces more tests while AssertFlip has a higher precision in test generation. On one hand, AssertFlip, with its verify and flip approach, already pre-verifies many test implementation details prior to reproduction test generation. This method means that AssertFlip is less likely generate tests that have *Mechanical Failure* or *Misimplementations* leading to an overall more precise set of generated tests at the cost of reduced generation volume. On the other hand, OpenHands, with its agentic approach, generates the tests directly using LLM without implementation details verification which led to a higher rate of *Mechanical Failure* due largely to incorrect environmental setup.

Overall, different approaches can have different effects on how precisely the tests are generated. Specifically, AssertFlip implementation details verification reduce the amount of generated tests but improve precision, whereas OpenHands's agentic approach favors coverage with more tests generated at the cost of lower precision.

> **Finding 2:** Different tools using different verification approaches can alter the precision and coverage tradeoff. By pre-verifying the implementation details, AsserFlip generates fewer tests than OpenHands (263 vs.464) but avoids more Mechanical Failures (7.6% vs 21.8%) and Misimplementations (8.0% vs. 15.1%).

## 6.3 RQ3: How Does a Model's Knowledge and Reasoning Ability Alter Failure Class Distributions?

Given that both tools heavily rely on LLMs to generate test cases, we seek to investigate the effect of model selection on the the distribution of failure classes. To this end, RQ3 will compare the frequency and proportion of observed failures between GPT-4o-mini (figure 1.b) and GPT-5-mini (figure 1.c) on AssertFlip.

Overall, the distribution changes significantly between GPT-4o-mini and GPT-5-mini, with GPT-4o-mini incurs a much bigger portion of *Mechanical Failures* (14.1% vs. 7.6%) and *Misimplementations* (25.4% vs. 8.0%). This difference indicates that GPT-4o-mini makes more mistakes due to its smaller knowledge base and weaker reasoning ability. GPT-5-mini is able to generate much more precise tests without *Mechanical Failure* and *Misimplementations*. Notably, *Wrong API Call* and *Incorrect Assertion* are the most prevalent failure modes in GPT-4o-mini generated tests. For example, in the case of django-15499, GPT-4o-mini generates a test which fails due to misunderstanding that optimization logic in django's reduce method is a multi-step sequence. But the tool attempted to verify it using a single comparison call while asserting that the result should contain the cumulative changes of two separate operations. Since the API only handles one peer-to-peer merger at a time, the second set of changes was never actually processed.

GPT-5-mini rectifies this test by utilizing the MigrationOptimizer class, which provides the correct high-level abstraction. By passing the entire list of operations into the optimizer, the test allows Django to handle the iterative reduction logic internally, accurately reflecting the desired outcome where multiple operations collapse into a single, updated CreateModel. Therefore, GPT-5-mini, unlike GPT-4o-mini, leverages its stronger knowledge base to interact with the correct API method to generate an accurate bug reproduction test.

Furthermore, GPT-5-mini was able to generate significantly more tests compared to GPT-4o-mini (263 vs. 71). This difference indicates that GPT-5-mini is much more capable at creating syntactically correct test which leads to both better precision and coverage.

> **Finding 3:** Models with a larger knowledge base and higher reasoning capability generate more tests with fewer failures. GPT-5-mini significantly outperforms GPT-4o-mini with more generated tests (263 vs. 71) with much lower failure rates (17.1% vs. 39.4%).

## 7 Discussion: Can the taxonomy provide insight for issue description enrichment that can improve test generation?

To determine if our taxonomy can give concrete insights to practitioners, we conduct a small experiment where the insights from our findings can guide augmentations that are applied directly to an issue's natural language description. This method, in turn, can help improve the outcome of the generated test. Table 3 presents our attempts with specific issues (Col *Issue*), in different subclasses (Col *Failure Subclass*). The *Issue Summary* column describes an issue's problem statement, and finally, we provide our concrete addition

to the issue description that can help improve the generated test quality. Specifically, for each failure mode, we craft an augmentation template that focuses the LLMs on avoiding the mistake in the first place. For this experiment, we utilize AssertFlip and GPT-5-mini.

**Wrong API Call:** For failures of this type, we focus the LLMs with the following template: *Ensure to fully understand and properly use the X libraries to the fullest extent necessary to correctly replicate the bug*, where X is the specific API(s) that the LLM needs to pay attention to. Table 3 shows the issues to which we apply this augmentation strategy. Of the six issues, two generate correct test cases after the augmentation. Specifically django-15382, django-13195, sympy-19495 and sympy-15976 fail to produce valid bug reproduction tests. Unlike the following failure categories, these samples rely upon a considerable understanding of the library they will be using to effectively replicate a bug. In each failing test, the mode of failure remained the same. This concentration of failures demonstrates how, library understanding requires a significant amount of context which the tool may not be able to accommodate. For example, in django-13195, the failure persists due to the model's inability to comprehend the complexities of the web-framework along with the necessary configuration for a web cookie. This failure can be contrasted by the success of django-13807. Whereas django-13195 failed due to insufficient API context, the subject matter of django-13807, generating a simple database table, was of reasonable complexity for the model to properly replicate. Therefore, failures within the Wrong API Call type occur due to a need for a deeper contextual understanding of the interacted libraries.

**Incorrect Input/Mock:** To focus the LLMs on making sure the input are generated or mocked correctly, we use the following template: *Ensure to fully, to the extent needed, understand and emulate the behaviour, input requirements and state of X*, where X is the specific input object or class that needs to be correctly initialized or mocked. We apply this augmentation strategy to pytest-5262 and xarray-6721, which are presented in Table 3. Of the two issues, both generate correct test cases after the augmentation. By providing a simple command to sufficiently mimic the behavior of their target mock, the tools replicated their respective issue's bug. This success demonstrates that, when given a minimal amount of context, the tool can create an effective mock, thus replicating the bug.

**Incorrect Assertion:** To ensure the tool generates accurate assertions, we use the following template: *Review your target output and the expected behaviour, make sure your assertions fully reflect the X*, where X is specific target which the assertion should focus on. We apply this augmentation strategy to astropy-14309, django-13212 and sphinx-10466 which are presented in Table 3. Of the three issues, both generate correct test cases after the augmentation. Therefore, the augmentation to review each issue's expected behaviuor and target output successfully resolved the failing instances. For instance, astropy-14309 resolves its index error by either checking for an empty list or a list of valid file formats. This is in contrast to the failing test which only asserted an empty list. Ultimately, this augmentation encourages the LLM to undertake a deeper review of the issue's requirements and methods of assessment.

Overall, our quick experiment shows a general success of context enrichment guided by our failure taxonomy with 7 out of 11 augmentations successfully improving the generated tests. This result shows that our taxonomy can be a concrete guide to users

**Table 3: An Overview of Candidates for Context-Enrichment**

| Issue | Failure Type | Issue Summary | Problem Statement Addition | Resolved Post-Augmentation? |
|---|---|---|---|---|
| django-13807 | Wrong API Call | loaddata crashes when table names are SQL keywords | ++Ensure to fully understand and properly use the **django.db.connections** library to the fullest extent necessary to correctly replicate the bug | yes |
| sympy-15875 | | A complex integer expression which evaluates to zero does not return true when is_zero() is called | ++ Ensure to fully understand and properly use the **sympy I and sympy simplify** libraries to the fullest extent necessary to correctly replicate the bug | yes |
| sympy-15976 | | A symbol ending with a number is invisible as the API wraps the expression in a `<mi>` tag | ++Ensure to fully understand and properly use the **sympy.printing.mathml and xml.dom** libraries to the fullest extent necessary to correctly replicate the bug | no |
| sympy-19495 | | the *.sub* method corrupts a bound variable | ++Ensure to fully understand and properly use the **ImageSet, ConditionSet and FiniteSet** libraries to the fullest extent necessary to correctly replicate the bug | no |
| django-13195 | | Deleting a cookie should preserve its samesite | ++Ensure to fully understand and properly use the **django.test.override_settings** library to the fullest extent necessary to correctly replicate the bug | no |
| django-15382 | | Django filter command removes WHERE command | ++Ensure to fully understand and properly use the **django.models.exists and base python** libraries to the fullest extent necessary to correctly replicate the bug | no |
| pytest-5262 | Incorrect Input/Mock | EncodedFile should not accept binary | ++Ensure to fully, to the extent needed, understand and emulate the behaviour, input requirements and state of **RawIOBase or BufferedIOBase interface**. | yes |
| xarray-6721 | | Accessing array chunks loads the whole array | ++ Ensure to fully, to the extent needed, understand and emulate the behaviour, input requirements and state of **xarray.Variable** | yes |
| astropy-14309 | Incorrect Assertion | Resolve IndexError | ++Review your target output and the expected behaviour, make sure your assertions fully reflect **all possible file formats that .ecsv should return** | yes |
| django-13212 | | ValidationErrors should include the provided value in a custom error message | ++Review your target output and the expected behaviour, make sure your assertions fully reflect **the population of the params dictionary of ValidationError** | yes |
| sphinx-10466 | | makeġettext returns duplicated list locations | ++Review your target output and the expected behaviour, make sure your assertions fully reflect **the Catalog as the sole source of truth** | yes |

in improving reproduction test generation using LLMs, and future research can utilize such a taxonomy to develop techniques that automatically improve the overall performance of LLM-generated reproduction tests.

## 8 Threats to Validity

**Data Leakage:** The risk of data leakage increases as models continually undergo training. As LLMs are generally trained on a large corpus of data, and the details of these training exercises are not released, the model can be trained on the issues within the SWT-Bench datasets. This form of leakage could introduce false positive results, as passing bug reproduction tests would be achieved via memorization rather than learning.

**LLM Stochasticity:** Given the probabilistic nature of LLMs, the exact results, even when using the same input parameters, may differ. Naturally, this threat may diminish the independent ability to verify this taxonomy through direct replication. Despite this threat, the taxonomy can be independently assessed via extension to other tools and models.

**Human Subjectivity:** While the labelling process followed a majority consensus model, the potential for Researcher Bias remains a threat. Individual experience with the Python ecosystem or specific

libraries may have influenced the classification of failure modes. The extension of an existing taxonomy, blind labelling and consensus methodology mitigate this threat by limiting interpretation and encouraging evidence-based justification, especially in the event of disagreement. Despite this mitigation technique, the act of labelling ensures that the threat of bias remains ever-present.

## 9 Conclusion

Given the importance of automated bug reproduction frameworks in improving program repair, understanding the failures of these tools is integral. This study addresses the existing literature gap by generating a taxonomy of automated bug reproduction test failures of state-of-the-art open-source frameworks using two modern GPT models Our findings indicate that different framework approaches can lead to different precision and coverage tradeoff and more powerful LLMs are more likely to increase the quality and quantity of generated tests. We also demonstrate that our taxonomy can be used as guidance to targeted augmentation which can lead to improvement of generated reproduction tests.

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
