# OpenReview forum: "A Taxonomy of Failing Bug Reproduction Tests on SWT-Bench"
_ACM.org/AIWare/2026/Conference — Submitted to AIware 2026_

### Official Review · Reviewer_9c7T · 2026-03-08

**Rating:** 1
**Confidence:** 5

**Review:**

Strengths
+ Clear motivation and gap.
+ Manual labeling effort and taxonomy construction.
+ Understanding why automated bug reproduction tools fail is interesting topic.

Weaknesses
- Poor reproducibility.
- Unsupported conclusions.
- Limited evidence of generalizability beyond the evaluated tools and models.
- The advertised evaluation matrix described in the abstract appears incomplete.

Detailed Comments:
- The abstract states that both frameworks (AssertFlip and OpenHands) are evaluated with GPT-5-mini and GPT-4o-mini. However, the results only report three configurations: AssertFlip + GPT-4o-mini, AssertFlip + GPT-5-mini, and OpenHands + GPT-5-mini. Results for OpenHands + GPT-4o-mini are missing with no justification.
- No released code or mapping from SWT-Bench instance IDs to taxonomy labels. This limits the ability to verify the taxonomy construction and reuse the labeled dataset.
- The paper presents its experimental design as controlled, but it is unclear which aspects were controlled: temperatures, prompts, token budgets etc.
- The paper attributes OpenHands + GPT-5-mini failures mainly to Environment Error, yet Table 2 shows that Wrong API Call occurs more frequently (46 vs 34). Even when considering the change relative to AssertFlip + GPT-5-mini, Wrong API Call increases more (+35 vs +29).
- The taxonomy is constructed concurrently with labeling but is not externally validated on new tools, models, or datasets. As a result, it is unclear whether the taxonomy would generalize beyond the studied configurations.
- The paper reports failure rates as percentages of generated tests (Figure 1), but later makes within-class claims (e.g., Incorrect Assertion representing about one third of misimplementations) using percentages normalized by generated tests. This creates confusion about the denominator used in the analysis. The distinction between these two normalization choices should be clearly explained.
- In RQ3, the paper attributes differences between GPT-4o-mini and GPT-5-mini to a “smaller knowledge base” and “weaker reasoning ability” but these attributes are not defined or measured in the study.
- The RQ3 analysis is incomplete. First, the comparison between models is performed only on AssertFlip. Second, the analysis focuses on proportions relative to the total number of generated tests, while examining distributions relative to the number of failed tests would also be important for answering the research question
- In Section 7, the evidence supporting taxonomy-guided context enrichment is based on only 11 samples, which the authors themselves describe as a “quick experiment”. This sample size is too small to support strong claims about the effectiveness of the proposed augmentation strategy.

**Summary:**

The paper studies LLM-based bug reproduction test generation on SWT-Bench. The authors evaluate two open-source frameworks, AssertFlip and OpenHands, under two model settings (i.e., GPT-5-mini and GPT-4o-mini). The main contribution is a manually constructed failure taxonomy. The authors label a set of failing instances, compare how failure types distribute across tool and model settings, and use these observations to answer three RQs about which types of failures are most common, how tool choice changes failure distributions, and how model capability changes these distributions. Finally, they run a small taxonomy-guided context enrichment experiment where issue descriptions are augmented based on the identified failure type, reporting that some previously failing cases can be fixed with targeted enrichment.

---

> ### Author Response · Authors · 2026-03-18
> **Reply to Reviewer 9c7T (1)**
>
> We appreciate the reviewer’s comments and will provide as much clarification as possible to the listed comments:
>
> **OpenHands-GPT-4o-mini**
> Our results for this model-tool pair are in the replication package (link is below). We will ensure to add the analysis to our paper. Specifically (with percentages displayed as relative to total generated tests for the model-tool pair), OpenHands-GPT-4o-mini displayed a higher proportion of misimplementations compared to OpenHands-GPT-5-mini (25.0% v 15.1%). When compared to AssertFlip-GPT-4o-mini, OpenHands-GPT-4o-mini had a much higher rate of mechanical failure (23.4% v 64.7%). This increase was driven by a higher proportion of Environment Error. Furthermore, the rate of misimplementations remains relatively constant across OpenHands-GPT-4o-mini and AssertFlip-GPT-4o-mini (25.0% v 25.3%).
>
> **Replication Package/Mapping to Issues**
> We originally planned to share the result files when the paper was accepted. We have prepared a public replication package which includes:
>
> - Our manual labelling sheet for all generated instances.
> - The failure traces and generated test code for all tool-model pairs.
>
> Please note that the preview feature displays on the first box below the “files” header. You can preview each file by selecting the appropriate preview button.
>
> Link: https://zenodo.org/records/19075731?token=eyJhbGciOiJIUzUxMiJ9.eyJpZCI6ImM4NWI5Y2MyLWVhOWUtNDJhYy1iYWM1LWE4ODA3Mzk4ZmRlNSIsImRhdGEiOnt9LCJyYW5kb20iOiJhZDVhNDA2OWQwYWFmMTljNjA5M2YzOWNmYmNiZmE0NiJ9.wRXbfMEcVoHBRn3DDtUvM9MjALIkk4RwRgXdenbY7sNC4ZNwrBO--nnU4jflBt2cGocRRy5aKmfm2HtIRbouHQ
>
> **Claim Regarding OpenHands GPT-5-mini and the Presence of Environment Error**
>  While Wrong API Call is the most frequent individual failure, the Mechanical Failure class (which includes Environment Errors) showed the most significant relative surge when moving from AssertFlip to OpenHands. For example, AssertFlip GPT-5-mini had a 1.9% observed rate of Environment Errors compared to 7.3% on OpenHands GPT5-mini. We will make sure to clarify this proportion in the paper.
>
>
> **Validation on Other Tools, Datasets, Models**
> The taxonomy categories are based on fundamental software engineering phenomena rather than occurrences which are specific to SWT-Bench. Whether a tool is agentic (like OpenHands) or a directed generator (like AssertFlip), it must still resolve these same challenges. Therefore, the taxonomy, capturing both agentic and rule-based approaches, should generalize to contemporary and future frameworks, models and datasets. We will add a section in our paper to clarify this.
>
> **Failure Rates as Percentages of Generated Tests**
> We appreciate the feedback and will ensure to keep the paper as clear as possible. Currently, all percentages use the total number of generated tests as the denominator to also demonstrate that the model-tool pairing generated passing tests along with failing tests. This comparison, whether using generated tests or failed tests would demonstrate the same trends given the common denominator. Where we discuss within-class distributions (e.g., Incorrect Assertions), we use words rather than percentages. To maximize clarity, we will explicitly state this in section 6.
>
> **RQ3 Analysis**
> We focused RQ3 on AssertFlip to isolate the model's performance from the tool's noise. The relative relationship between the failure classes would be the same if assessed between the generated tests and failed tests, since they'd have a common denominator. We believe presenting the percentages in text as a rate of total generated tests may better encompass the tools’ generation of passing and failing tests. Table 2 includes a breakdown of the failed and generated tests to provide an option to analyze the data on the basis of generated or failing tests.
>
> **Relevance of Taxonomy-Guided Prompt Augmentation**
> This discussion topic provides a small example to demonstrate the taxonomy’s potential for test improvement. While the experiment is small in scale, it provides a motivation for a broader analysis of its usefulness. By showing that even a simple augmentation based on our taxonomy can reduce Mechanical Failure and Misimplementation, we provide an empirical motivation for future research in incorporating the taxonomy to improve test generation tools.

---

> > ### Author Response · Authors · 2026-03-18
> > **Reply to Reviewer 9c7T (2)**
> >
> > **GPT-4o-mini, GPT-5-mini  “smaller knowledge base” and “weaker reasoning ability” Claims**
> >
> > We agree that the terms require stronger grounding, although direct measurement is beyond the scope of the current study. The GPT-5 documentation describes the capability differences between GPT-4o-mini and GPT-5-mini [1]. Furthermore, the knowledge base claim is supported by GPT-4o-mini’s knowledge cutoff of October 1, 2023 compared to May 31, 2024 for GPT-5-mini [2][3]. Additionally, independent benchmark results from LoCoBench compare the two models across difficulty levels with GPT-5-mini outperforming GPT-4o-mini on long-context reasoning, with the gap widening at Expert difficulty, consistent with a reasoning capability differential [4]. We will include these references to better ground the claim.
> >
> >
> > **Proposed Revision List**
> > - Clarify the Environment Error proportion on AssertFlip-GPT-5-mini v OpenHands-GPT-5-mini
> > - Describe our taxonomy’s generalizability
> > - Explicitly state that our percentages are relative to generated tests
> > - Provide clarity on OpenHands-GPT-4o-mini
> >
> >
> > **References**
> > [1] Jielin Qiu, Zuxin Liu, Zhiwei Liu, Rithesh Murthy, Jianguo Zhang, Haolin Chen, Shiyu Wang, Ming Zhu, Liangwei Yang, Juntao Tan, Zhepeng Cen, Cheng Qian, Shelby Heinecke, Weiran Yao, Silvio Savarese, Caiming Xiong, and Huan Wang. 2025. LoCoBench: A Benchmark for Long-Context Large Language Models in Complex Software Engineering. arXiv:2509.09614 [cs.SE]. Retrieved from https://arxiv.org/abs/2509.09614
> > [2] OpenAI. 2025. Introducing GPT-5 for Developers. OpenAI. August 7, 2025. Retrieved March 17, 2026 from https://openai.com/index/introducing-gpt-5-for-developers/
> > [3] OpenAI. 2025. GPT-4o mini. OpenAI API Documentation. Retrieved March 17, 2026 from https://developers.openai.com/api/docs/models/gpt-4o-mini
> > [4] OpenAI. 2025. GPT-5 mini. OpenAI API Documentation. Retrieved March 17, 2026 from https://developers.openai.com/api/docs/models/gpt-5-mini

---

### Official Review · Reviewer_GP4x · 2026-03-10

**Rating:** 2
**Confidence:** 4

**Review:**

**Strengths**
=============

*   The research motivation is clear, the paper addresses an important gap in understanding why LLM-based bug reproduction tools fail, which is relevant for improving automated program repair pipelines.

*   The study analyzes 254 failing reproduction tests, providing concrete empirical evidence about common failure modes and a structured categorization of bug reproduction failures.

*   The taxonomy is used to guide issue description augmentation, demonstrating a potential pathway for improving LLM-generated tests.


**Limitations**
===============

*   The study relies only on two bug reproduction frameworks and exclusively on SWT-Bench, and it is unclear whether the taxonomy would generalize to other reproduction test benchmarks.

*   The selected two LLMs represent only a small subset of bug reproduction frameworks, excluding other LLM-based test generation approaches.

*   The evaluation of the taxonomy usefulness is limited**.** The paper demonstrates taxonomy-guided improvements using a very small experiment (11 cases), which is insufficient to draw strong conclusions about the taxonomy’s practical impact.


**Detailed Comments**
=====================

*   The paper addresses a timely and relevant problem in LLM-based software engineering tools by focusing on understanding failure modes in automated bug reproduction frameworks

*   While the taxonomy is useful for understanding failure patterns, the experimental scope is relatively limited, as the analysis includes only two frameworks and two LLM models. The work could be strengthened by evaluating the taxonomy on additional benchmarks, tools, or LLM architectures to assess its general applicability

*   Since the taxonomy requires manual labeling of failures, the paper could further explore methods for automatically detecting failure categories from test execution traces or model outputs.

*   The taxonomy-guided prompt augmentation experiment is interesting but very small in scale, making it difficult to assess the broader impact of this approach.


**Questions for the Authors**
=============================

**1-** How well do you expect the proposed taxonomy to generalize to other bug reproduction frameworks or LLM-based testing tools beyond AssertFlip and OpenHands?

**2-** Since the study relies entirely on SWT-Bench, have you considered evaluating the taxonomy on other reproduction benchmarks such as GitBugJava or CrashJS?

**3-** How does your taxonomy compare with existing failure taxonomies in software testing or LLM-generated code evaluation? Have you considered empirical comparisons?

**Summary:**

The paper presents a taxonomy of failure modes in LLM-based bug reproduction test generation frameworks using the SWT-Bench benchmark. The authors analyze failures produced by two open-source bug reproduction tools, AssertFlip and OpenHands, using two language models (GPT-4o-mini and GPT-5-mini). Results show that Mechanical Failures and Misimplementations are the most common failure categories, while stronger models generate more tests with fewer errors. The authors also conduct a small experiment showing that taxonomy-guided prompt augmentation can improve reproduction test generation in some cases.

---

> ### Author Response · Authors · 2026-03-18
> **Reply to Reviewer GP4x**
>
> We appreciate the reviewer’s comments and will provide as much clarification as possible to the listed comments:
>
> **Extensibility to Other Frameworks**
> The taxonomy categories are based on software engineering phenomena rather than occurrences, which are specific to SWT-Bench. Whether a tool is agentic (like OpenHands) or a directed generator (like AssertFlip), it must still resolve these same challenges. Therefore, the taxonomy, capturing both agentic and rule-based approaches, should generalize to other frameworks. We will add a section in our paper to clarify this.
>
> **Evaluation on Other Benchmarks**
> We selected SWT-Bench for its popularity among several agentic tools, a provided leaderboard to track the most performant tools and its high-level scope. We did not focus on CrashJS or GitBugJava, as both benchmarks lack a standardized and containerized execution environment. Developing this environment from scratch for these benchmarks would be a significant undertaking in and of itself. The environment, like the one SWT-Bench provides, ensures all evaluation attempts are consistent, thus reducing a potential source of error within a paper’s evaluations. Despite the limitations, extensions to other benchmarks would be an interesting future work.
>
> **Comparison to Other Failure Taxonomies**
> We will edit our Related Work section to highlight our taxonomy’s difference to the taxonomy of Taherkhani et al. Our taxonomy provides added granularity through a hierarchical structure. Crucially, our labels capture out-of-execution failures that are specific to autonomous agents and invisible to traditional test-code-only taxonomies. Furthermore, our taxonomy is specific to bug reproduction tests whereas the taxonomy of Taherkhani et al pertains to LLM-generated tests [31].
>
> **Relevance of Taxonomy-Guided Prompt Augmentation**
> This discussion topic provides a small example to demonstrate the taxonomy’s potential for test improvement. While the experiment is small in scale, it provides a motivation for a broader analysis of its usefulness. By showing that even a simple augmentation based on our taxonomy can reduce Mechanical Failure and Misimplementation, we provide an empirical motivation for future research in incorporating the taxonomy to improve test generation tools.
>
> **Proposed Revision List**
> - Clarify the taxonomy’s extensibility to other frameworks and our similarity to other taxonomies
> - Clarify our sole selection of SWT Bench

---

### Official Review · Reviewer_wvCW · 2026-03-11

**Rating:** 2
**Confidence:** 3

**Review:**

Strengths:
+ An interesting and timely topic in automated program repair (APR).
+ The paper is generally easy to read and follow.

Weaknesses:
- The methodology could be better grounded and more rigorously justified.
- The presentation of the study results, particularly the taxonomy/categories, could be improved.
- A replication package should be provided to strengthen the validity and reproducibility of the study.

Detaild Comments:

- Organization: The paper could be better organized. For example, an organization paragraph outlining the structure of the paper should be included at the end of the Introduction section.

- Supporting References: Certain statements require stronger support. For example, the claim that "AssertFlip and OpenHands represent the top two open-source frameworks on the SWT-Bench Verified leaderboard" should be supported with a citation or empirical evidence.

- Study Subjects: Why does the study focus only on the GPT model series? The rationale for selecting the GPT models should be clearly explained.

- Taxonomy Construction: The paper states that "Two authors began the process by agreeing on an initial set of categories and independently labelling all failing instances from our selected frameworks and LLMs". However, how was the initial set of categories derived? Which qualitative data analysis method (e.g., open coding, grounded theory, thematic analysis) was employed? How was Cohen's kappa calculated to measure inter-rater reliability for the categorization process? These methodological details are essential for ensuring the methodology rigor and transparency.

- Relation to Existing Taxonomies: How do the proposed categories overlap with or differ from existing taxonomies? This relationship should be explicitly discussed.

- Results: Table 2 presents the number of failed tests in each failure class. Additional explanation and interpretation would help readers better understand the significance and implications of these results.

- Hypotheses: In the end of Section 6.1, the authors state that they "will explore hypotheses further in Section 7", but it is not clear which specific hypotheses are being examined. These should be explicitly stated and clearly connected to the results.

- Replication Package: The authors are encouraged to provide a publicly available replication package so that the community can validate and replicate the study.

Minor issues:
- table 2 -> Table 2 (capitalize Table)
- The text in Figure 1 is difficult to read and should be improved for clarity.

**Summary:**

This paper investigated the limitations of automated bug reproduction test generation for Automatic Program Repair (APR), focusing on why state-of-the-art frameworks still struggle with complex issues. The authors proposed a taxonomy of test generation failures by analyzing error traces from two leading open-source tools, AssertFlip and OpenHands, evaluated on the SWT-Bench Verified benchmark. The study further examined the impact of model selection by employing GPT-5-mini and GPT-4o-mini, revealing that both the generation approach and model capability significantly influence test precision and coverage. Their analysis identifies mechanical and implementation issues as the dominant causes of failure, and the results show that more advanced models with stronger reasoning abilities exhibit fewer context-related failures. The authors demonstrated that the proposed taxonomy can guide the augmentation of issue descriptions to improve test generation quality. Overall, the paper provides an empirical characterization of failure modes in bug reproduction testing and offers practical insights for enhancing APR pipelines.

---

> ### Author Response · Authors · 2026-03-18
> **Reply to Reviewer wvCW**
>
> We appreciate the reviewer’s comments and will provide as much clarification as possible to the listed comments:
>
> **Paper Organization**
> We agree that an explicit roadmap helps the reader. We will add a paragraph at the end of the Introduction outlining the structure of the paper
>
> **Supporting References**
> Our paper is structured such that all claims, on their first invocation, are grounded with a reference or value. For example, the claim that AssertFlip and OpenHands represent the top two open source frameworks is first introduced and grounded with a reference in the second paragraph of the introduction. We will update the paper to ground all invocations with a citation
>
> **Selection of GPT Series Models**
>
> The choice of the GPT family was driven by the current SOTA in bug reproduction. OpenHands and AssertFlip respectively use GPT-5 and GPT-4 class models for their top-performing runs on SWE-bench. We selected the mini variants of these models as cost-effective, high-performing representatives of this family. We will clarify this selection in our paper.
>
> **Taxonomy Category Creation and Agreement Calculation**
>
> Our taxonomy was created by thematic analysis. We were inspired by the work of Taherkhani et al., but did not adapt the taxonomy as their work pertained to a different subject. Specifically, their work pertains to LLM-generated test cases, whereas we focus on bug reproduction tests. Therefore, our taxonomy must better encompass issues at a repository level. We transitioned to a thematic analysis phase where two authors independently grouped observations into overarching themes (Mechanical, Misimplementation, Requirement Misunderstanding), resulting in our proposed hierarchical structure. We will clarify the use of thematic analysis in the paper and describe the steps to match Braun and Clarke’s structure.
>
> Cohen’s Kappa was measured between the two labelling authors’ agreement on the labelled sets. To ensure the paper remains self-contained, we will include both explanations for the in-camera version.
>
>
> **Relation to Existing Taxonomies**
> To our knowledge, there is no taxonomy for bug reproduction test failures. The closest taxonomy that we are aware of is that of Taherkani et al. Our taxonomy design focuses on phenomena which are general to the software engineering domain rather than failure types which may present itself more frequently in a specific domain (IE failures specific to a model, format, tool, dataset). Given this feature and our subject matter, our taxonomy should be more widely applicable.
>
> **Interpretation of Table 2**
> The explanation of the values within table two are explained in the following paragraphs after its introduction at line 452.  This presentation is meant to provide a raw depiction of the taxonomy construction to complement the normalized depiction in Figure 1. We will ensure to clarify the explanation.
>
> **Hypothesis in Section 7**
> At line 601, we state the hypothesis as, “This finding suggests that most issues
> fail as a result of a lack of logical or contextual clarity rather than
> high logical complexity, and further clarification in the issue text
> would help the model generate more correct test cases”. We will reiterate the hypothesis in Section 7 to increase clarity.
>
> **Replication Package/Mapping to Issues**
>
> We originally planned to share the result files when the paper was accepted. We have prepared a public replication package which includes:
>
> Our manual labelling sheet for all generated instances.
> The failure traces and generated test code for all tool-model pairs.
>
> Please note that the preview feature displays in the first box below the “files” header. You can preview each file by selecting the appropriate preview button.
>
> Link: https://zenodo.org/records/19075731?token=eyJhbGciOiJIUzUxMiJ9.eyJpZCI6ImM4NWI5Y2MyLWVhOWUtNDJhYy1iYWM1LWE4ODA3Mzk4ZmRlNSIsImRhdGEiOnt9LCJyYW5kb20iOiJhZDVhNDA2OWQwYWFmMTljNjA5M2YzOWNmYmNiZmE0NiJ9.wRXbfMEcVoHBRn3DDtUvM9MjALIkk4RwRgXdenbY7sNC4ZNwrBO--nnU4jflBt2cGocRRy5aKmfm2HtIRbouHQ
>
>
> **Minor Text Issues**
> Thank you for pointing this out. We will implement these suggestions for the in-camera version
>
> **Proposed Revision List**
> - Add a paragraph to outline the structure of the paper
> - Ground all references to a claim with a citation rather than citing at a claim’s first invocation
> - Clarify the selection of the GPT series of models
> - Clarify the taxonomy construction process using Thematic Analysis
> - Describe Cohen’s Kappa
> - Clarify the complementary nature of table 2 and figure 1
> - Clarify the hypothesis mentioned in 6.1

---

> > ### Comment · Reviewer_wvCW · 2026-03-18
> >
> > Thank the authors for answering the comments. I have one remaining question:
> >
> > **Taxonomy Category Creation and Agreement Calculation**: The authors state that "two authors began the process by agreeing on an initial set of categories and independently labeling all failing instances from the selected frameworks and LLMs", What are the initial set of categories? In cases where some instances do not fit into the initial set of categories, how were such instances handled? Additionally, how was Cohen’s kappa calculated under these conditions?

---

> > > ### Author Response · Authors · 2026-03-19
> > >
> > > We thank the reviewer for their question:
> > >
> > > **Taxonomy Category Creation and Agreement Calculation**
> > >
> > > The initial set of categories was: Output Format Inconsistency, Wrong API Call, Incorrect Input/Mock, Logical Failure, Misunderstanding Edge Case Logic, and Misunderstanding Function Logic From Natural Language. When instances did not fit into the original categories, they were flagged for the first and second authors to review, generating new categories with the third author being consulted if consensus cannot be reached. The authors would then independently label the outstanding instances. Cohen’s Kappa was calculated at the end of the process.

---

> > > > ### Comment · Reviewer_wvCW · 2026-03-19
> > > >
> > > > This process does not appear to reflect fully independent labeling. As stated “when instances did not fit into the original categories, they were flagged for review by the first and second authors, with new categories generated and the third author consulted if consensus could not be reached”. This suggests an iterative and collaborative coding process rather than strictly independent annotation. Therefore, I am not fully convinced that Cohen's kappa is an appropriate measure in this context, as it assumes a fixed coding scheme and independent labeling.

---

> > > > > ### Author Response · Authors · 2026-03-20
> > > > >
> > > > > We thank the reviewer for their comment:
> > > > >
> > > > > To clarify, the labeling is done independently. The iterative discussion to reach consensus is only needed for 4 new categories. This creation is distinct from assigning a label to an instance. These labels were applied 189 times on the data (434 total instances including OpenHands GPT-4o-mini). Thus, *labelling was still conducted independently*. Therefore, Cohen’s Kappa, which is calculated on the labeling agreement, would still reflect the consistency of the final assigned label and indicate a clear coding scheme. We will make sure to clarify this in the paper.

---

### Author Response · Authors · 2026-03-18

We thank the reviewers for their comments and time. We have provided clarification to each reviewer’s comments using the Official Comment feature for each review. Our planned revisions include the following:

- Add a paragraph to outline the structure of the paper
- Ground all references to a claim with a citation rather than citing at a claim’s first invocation
- Clarify the selection of the GPT series of models
- Clarify the taxonomy construction process using Thematic Analysis
- Describe Cohen’s Kappa
- Clarify the complementary nature of Table 2 and Figure 1
- Clarify the hypothesis mentioned in 6.1
- Clarify the taxonomy’s extensibility to other frameworks and our similarity to other taxonomies
- Clarify our sole selection of SWT Bench
- Clarify the Environment Error proportion on AssertFlip-GPT-5-mini v OpenHands-GPT-5-mini
- Describe our taxonomy’s generalizability
- Explicitly state that our percentages are relative to generated tests
- Provide clarity on OpenHands-GPT-4o-mini